# Spatio-temporal variation of Cerambycidae-host tree interaction networks

**Michelle Ramos-Robles**[¤], **Orthon Ricardo Vargas-Cardoso, Angélica María Corona-López, Alejandro Flores-Palacios, Víctor Hugo Toledo-Hernández***

Universidad Autónoma del Estado de Morelos, Centro de Investigación en Biodiversidad y Conservación, Chamilpa, Cuernavaca, Morelos, México

¤ Departamento de Botánica, Universidad Autónoma Agraria Antonio Narro, Buenavista, Saltillo, Coahuila, México

* victor.toledo@uaem.mx

**Data Availability Statement:** All relevant data are within the manuscript and its Supporting Information files.

## Abstract

Despite its high ecological importance, the commensal interactions at community level are poorly studied. In tropical dry forests (TDF) there is a great diversity of species adapted to the high seasonality that characterizes them; however, little is known regarding how the spatial and temporal availability of resources generates changes in the pattern of commensal interactions. We experimentally studied changes in the diversity, composition, and pattern of interactions in spatio-temporal associations between the saproxylophagous beetles and their host trees in a TDF in Morelos, Mexico. A total of 65 host tree species were selected, from which 16 wood sections were obtained per species. These sections were exposed in the field to allow oviposition by the cerambycids under four different (spatio-temporal) treatments. We analyzed the network structure and generated indices at species level (i.e., specialization, species strength, and effective partners) and those related to physical characteristics of the wood (hardness and degradation rate) and the cerambycids (body size). In total, 1,323 individuals of 57 species of cerambycids emerged. Our results showed that, independently of the space and time, the network presented a nested and modular structure, with a high specialization degree and a high turnover of cerambycid species and their interactions. In general, we found that the cerambycids are mostly associated with softwood species with a lower decomposition rate of wood, as well as with the most abundant host species. The commensalistic interactions between the cerambycids and their host trees are highly specialized but are not spatio-temporally static. The high turnover in the interactions is caused by the emergence patterns of cerambycids, which seem to restrict their use to certain species. The knowledge of the spatio-temporal variation in Cerambycidae-host tree interactions allows us to predict how environmental and structural changes in the habitat can modify the species ensemble, and therefore its interactions.

## Introduction

The tropical dry forest (TDF) is an ecosystem with high biodiversity [1] and the species that inhabit it are adapted to survive under seasonal conditions of scarce rain [2]. These forests

**Funding:** This research was funded by a Consejo Nacional de Ciencia y Tecnología (CONACyT) (grant 274685) awarded to ORVC, and by the post-doctoral Programa para el Desarrollo Profesional Docente PRODEP (grant 511-6/17-10976) awarded to MRR. The funders had no role in study design, data collection and analysis, decision to publish, or preparation of the manuscript.

**Competing interests:** The authors have declared that no competing interests exist.

develop in highly seasonal climates, which generates abiotic conditions that are more stressful than those of the tropical rain forests [3]. The climatic seasonality of the TDF determines the variation in the spatial and temporal availability of the resources (e.g., food and water) and thus generates changes in the abundance, richness, composition, and interactions of the species [4, 5]. Part of the biota of the TDF can survive the drought because they present resting cycles below ground (e.g., hemicryptophyte and cryptophyte plants), or remain active in chambers that provide them with suitable conditions to continue growth, such as in the case of the saproxylophagous insects. These insects spend most of their lives as larvae within the trees, in chambers where they eat, grow and are protected from the exterior conditions. Their life as an imago is shorter than that as a larva and they emerge only to reproduce and die [6].

The spatio-temporal variation of biodiversity has been documented in several studies [7–10]. The canopy has been recognized as an important pool of insect diversity [11] and the presence of a vertical stratification of insects has been shown throughout the canopy of the tropical forests [5, 12, 13], with greater, lower or equal abundances and diversities found among the strata [12, 13]. These differences may be due to sampling type, forest structure (which determines resource availability and microclimate) or to interactions with competitors and predators [14–16]. The insects associated with a vertical stratum must be adapted to changes in light, humidity, and temperature, in contrast to insects associated with the soil, where the conditions of humidity are greater and more constant compared with the canopy [17]. However, studies of vertical stratification to date have focused on searching for adult activity patterns and not in experimental determination of whether or not selection of the substrate is conducted by the adults for development of the larvae [18, 19] but see [20].

Within the group of insects, at least one-third of species belong to the guild of the saproxylophages (i.e., insects that in some part of their life cycle depend on dead or dying wood; [21]). This guild includes the Longhorn beetles (Cerambycidae), which are the first beetles to colonize dead wood (i.e., early successional beetles), facilitating the subsequent entry of other saproxylophagous insects of late succession [22]. The larvae of this group possess morphological and physiological modifications that enable them to feed on the wood of fallen, diseased, dead or recently felled trees [23, 24]. However, there is a high content of defensive compounds in the tissues of these woods, for which reason it has been hypothesized that some species of cerambycids show specialization in their host use [25]. Because the cerambycids are involved in the process of decomposition of wood and recycling of nutrients, they have been considered a bioindicator group [26–28]. However, despite their importance, there are no studies that experimentally test the specialization in the use of wood by cerambycids and the spatio-temporal variation that exists in these preferences.

The complex networks method has been used to test hypotheses and to understand how ecosystems are structured through the interactions between communities; however, its use has been biased towards systems of mutualistic and antagonistic interactions [29, 30], so there is a deficit of studies about commensalistic interactions (but see [31]). In general, network analysis has shown that, within an ecosystem, interactions occur in a non-random manner and generate a network of interactions with a structure (e.g., nested, modular or both), suggesting relationships among the interacting species that could be the result of ecological (e.g., competition among species of the same guild) or evolutionary mechanisms (e.g., coevolution among communities; [32]). In addition, it is possible to test hypotheses regarding how the characteristics of plant (e.g., height, diameter at breast height, chemical defenses; [33, 34]) or animal species (e.g., body size, morphological features; [35, 36]) explain the structure of a network, or to evaluate the species role in the network [37], due to the number of interactions, level of specialization (selectivity) or their value of species strength (e.g., indicates that host tree species are relatively important to the cerambycids).

In commensalistic interactions, a group of species (e.g., trees) may not have a cost because their dead parts are used, while the other group (e.g., cerambycids) is benefited. So, it can be expected that cerambycids make general use of trees, generating low nested, modular and specialized networks. In recent years some studies have analyzed the Cerambycidae-host tree commensalistic interactions through the network's method, finding that, as with mutualistic and antagonistic interactions, these present a defined structure that is mainly nested (e.g., [38–40]. However, they present lower values of nestedness than other networks of commensalistic interactions (e.g., epiphyte-phorophyte NODF = 59.44 ± 20; [41]), showing high specialization level in the early stages of wood decomposition ([25, 40]. Wende et al. [25] showed that these levels of specialization are related to the characteristics of the plants (i.e., their chemical compounds) and animals (i.e., body size). So, although it is possible to hypothesize that the relationships between cerambycids and their trees are generalists, few studies suggest that the generality in the use of wood is restricted by the characteristics of the wood and the body size of the insects [25, 40]. Despite the progress in the detection of patterns and characterization of the Cerambycidae-host tree commensalistic interactions, it is not known how these interactions vary spatio-temporally and how the physical characteristics of the wood and of the cerambycids will be related to the network parameters, particularly in highly seasonal habitats such as the TDF.

We experimentally studied spatial (canopy vs. ground) and temporal (dry vs. rainy) associations between the saproxylophagous beetles of the family Cerambycidae and their host trees in a TDF and hypothesized that:

1. Regardless of space-time, the Cerambycidae-host tree commensalistic network will be specialized, for which reason it will present a lower niche overlap among the species of saproxylophages, due to the physico-chemical limitations of the wood in early stages of decomposition in terms of consumption by the saproxylophages.

2. The diversity and composition of species, as well as the interactions turnover of cerambycids, will change as a result of the highly seasonality of the TDF, being more adverse in the dry season and in the canopy, in which there is limited decomposition of wood due to the reduced availability of water.

3. The size of the cerambycids species will be positively related to specialization at species level, due that the larger species are less abundant and have more specific energetic requirements.

4. The importance of host species will be inversely related to the hardness and decomposition rate of the wood because softer woods decompose faster and can be used in an environment where the rainy season is short.

## Materials and methods

### Ethics statement

Insects used in this work were collected and treated with standard ethical procedures [42]. The field study was approved by the authorities of the Municipal assistant of San Andrés de la Cal, granted access to the communal property of the studied area and the collection of specimens by national authorities of Secretary of Environment and Natural Resources (Secretaría de Medio Ambiente y Recursos Naturales; SEMARNAT, permission FAUT-0178). According to Mexican Official Standard NOM-059-SEMARNAT-2010, the trees *Sideroxylon capiri* and *Sapium macrocarpum* are protected species in the studied area, the latter is the dominant tree

species in several forest patches and for all the studied tree species we only took branches, without compromising the survival of the individual, following standard ethical pruning techniques [43].

## Study area

This study was conducted in the strongly seasonal tropical dry forest (TDF) of San Andrés de la Cal, Tepoztlán, Morelos, Mexico (19˚00'N and 99˚05'W), at an elevation of 1500 m a.s.l. The mean annual temperature is 20˚C, with a mean annual precipitation of 1200 mm [44]. This forest is located in the lowest part of the protected natural area known as Chichinautzin Biological Corridor. The TDF has an open canopy with short trees (maximum of 16 m in height). The dominant tree species are *Sapium macrocarpum* Müll. Arg. (Euphorbiceae), *Bursera fagaroides* (Kunth) Engl., *B. glabrifolia* (Kunth) Engl., *B. copallifera* (DC.) Bullock (Burseraceae), *Ipomoea pauciflora* M. Martens & Galeotti (Convolvulaceae) and *Quercus obtusata* Bonpl. (Fagaceae) [45, 46].

**Interaction sampling of Cerambycidae beetles.** To record interactions between the saproxylophagous beetles and their host tree species, a field experiment was conducted. A total of 65 host tree species were selected based on previous studies of cerambycid specificity [47]. The selected individuals were of diameter at breast height (DBH) $\geq$ 2 cm, and height $\geq$ 2 m. Rearing of cerambycid larvae was chosen as the most suitable method for determination of the preferences of this group for their host trees [20].

From February 2015 to March 2016, branches that appeared visibly healthy were cut from each host tree and divided into 16 sections of 50–60 cm in length. In order to determine the spatio-temporal patterns of the Cerambycidae-host tree interactions, each branch of each individual was assigned to one of four different treatments: during the rainy season, four branches were left in the canopy (Rain-canopy, Rc) and four on the ground (Rain-ground, Rg), during the dry season, four branches were left in the canopy (Dry-canopy, Dc) and four on the ground (Dry-ground, Dg). Due to the differential abundance of the species and the destructive nature of the experiment that restricted us to using large trees with a sufficient number of branches, the number of individuals selected per tree species varied from 1 to 5.

In order to allow oviposition by the cerambycids, the sections of the branch were left exposed in the TDF and, every two months, two sections from each individual were collected (four collection intervals: 2, 4, 6 and 8 months). These were then taken to the laboratory where each branch section was isolated with a galvanized wire mesh and the number of cerambycid individuals that emerged from each section of branch quantified.

**Species traits.** To determine which characteristics of the wood were related to the pattern of interactions of the cerambycids, the hardness and the percentage of degradation of the wood was estimated in each host. We used as a proxy of hardness the wood density. To measure wood density, from May 2016 to September 2017, another four branches of 3–4 cm in diameter and 10 cm length were collected from the 65 species where cerambycids emerged. Wood density was determined using the following formula:

$$Wood\ density = \frac{branch\ dry\ weight}{dry\ branch\ volume}$$

To obtain the dry weight of each section of branch, the sections were placed in a drying oven (Binder® FD 115-UL, Germany) at 100˚C for three days, until reaching a constant weight, which was taken as the dry weight [48]. The volume of the dry branches was measured by water displacement [48], using a graduated cylinder (1000 mL).

As with the measurement of wood hardness (see above), four branches of diameter 3 to 4 cm and length 50 to 60 cm were obtained in order to quantify the percentage of decomposition

of the branches. Their initial and final (at eight months) weights were recorded and the days of exposure of the branches quantified. With this information, it was possible to determine the percentage of daily decomposition (PDD) of the wood per host species using the following formula:

$$PDD = \frac{100 \times \left(1 - \frac{Initial\ weight - Final\ weight}{Initial\ weight}\right)}{Total\ days\ of\ exposure\ of\ branches}$$

As a functional characteristic, a body size index (BSI) of the cerambycids was quantified like a proxy of the energetic requirements of each species in order to complete its life cycle. To obtain this index of each beetle species, we photographed from 1 to 10 individuals depending on availability and used the program ImageJ 1.52a [49] to determine the length and width of each individual. With this information, we calculated a BSI by multiplying the values of length and width. Where there was more than one individual per species (44 species from a total of 57), we obtained the average BSI.

## Data analysis

**Diversity and composition of Cerambycidae beetles.** To compare the α diversity among the four treatments (Rc, Rg, Dc, and Dg), we calculated the effective number of species or Hill numbers ($^{q}D$) that quantifies the diversity in units of equivalent numbers of species on weighting the abundance with the value of q (q order).

We calculated $^{0}D$, which indicates the species richness; $^{1}D$ (exponential of the Shannon index), which indicates the number of equally abundant species; and $^{2}D$ (inverse of Simpson index), which indicates the number of equally dominant species of the community [50]. The three orders of diversity were compared among the treatments (Rc, Rg, Dc, and Dg), with the respective 95% confidence intervals (100 bootstraps) calculated with the package iNEXT [51].

To evaluate changes in the species composition of Cerambycidae beetles and their host trees among the different treatments (Rc, Rg, Dc, and Dg), we used the beta diversity partitioning method proposed by Baselga [52]. The Sorensen dissimilarity index ($β_{sor}$) was calculated. This index explains the total variation per treatment, including turnover ($β_{sim}$) and nestedness ($β_{sne}$). $β_{sim}$ indicates the changes in species composition as a result of the turnover of species. $β_{sne}$ represents the nestedness (i.e., the loss and/or gain of species) resulting from the difference between $β_{sor}$ and $β_{sim}$ [52]. The metrics were calculated with the package Betapart [53].

We calculated the dissimilarity of the network (or β-diversity) of interactions between each treatment based on Poisot et al. [54] in order to understand the spatio-temporal changes of the Cerambycidae-host tree interaction networks. We explored whether the differences among the treatment networks were due to two components: differences in species composition ($β_{ST}$, dissimilarity in the interactions due to the similarity in species composition), or because the shared species interact differently in each network ($β_{OS}$, similarity of interactions in species that co-occur; i.e., species rewiring). The analysis was conducted with the package Betalink [54].

**Network level.** A network of interactions was generated for each one of the treatments (Rc, Rg, Dc, and Dg). Nestedness and modularity were analyzed in each network. Nestedness quantifies the degree to which longhorn beetle with few interactions are connected to highly connected interactions species [55]. The nestedness was analyzed using the WNODF index (Weighted NODF, [56]). This index takes values between 0 (not nested) and 100 (perfectly nested). Modularity quantify the degree the interactions are structured in modules (i.e., sets of species that interact more intensely among themselves than with the rest of the community; [57]). Modularity index (Q) was calculated with the DIRTLPA+ algorithm for weighted

networks proposed by [58]. Q takes values between 0 and 1, with greater modularity indicated with proximity to 1 and vice versa [57].

For each network, we measured the specialization ($H_2'$) and niche overlap. $H_2'$ is a quantitative measure of specialization that controls the frequencies of interaction expected of the total observations per species. A value of 0 indicates no specialization, while 1 indicates perfect specialization [59]. In addition, we calculated the niche overlap for the beetles and hosts. This is a measure of the similarity of resource use among species of the same trophic level, in which values close to 0 indicate no common resource use and those close to 1 indicate a perfect superimposition of resource use [60]. We used a restrictive Patefield null model based on algorithm *r2dtable* which maintain fixes the total number of species interactions [61] with 1000 randomizations to analyze statistics significance of network metrics. Particularly, p values were calculated for WNODF, $H_2'$ and niche overlap and Q index values were used to calculate the *Z*-score (values $\geq$ 2 are modular) [57].

**Species level.** To determine the selectivity of the beetles for their host trees, we calculated the index of specialization (d') at species level. This index ranges from 0 to 1, with the highest values indicating greater selectivity [59]. The number of effective partners indicates the effective number of interacting species, assuming that each species is equally common, in a similar way to the value of true diversity $^1D$ [50]. Species strength was calculated as the sum of the values of the intensity of the interaction (interaction strength) of a beetle species with all of the host trees [62]. This parameter indicates what proportion of all of the interactions observed for a given species is produced with each species of the community [62]. High values of species strength indicate which species of host trees are relatively important for the cerambycids. All network analysis were performed with the package bipartite [63].

**Trait-based predictors of network parameters.** To determine how the physical characteristics of the wood of the host tree species and the saproxylophagous beetles relate to the parameters of each network (Rc, Rg, Dc, and Dg), we used a principal component analysis (PCA). In this analysis, the variables of the tree species were the hardness and decomposition rate of the wood. For the beetles, the mean BSI and, for both groups, the indices of the network, with respect to each trophic level (degree, d', effective partners, species strength). All of the PCA were based on a matrix of correlations; for interpretation of the eigenvectors (principal component, PC) we considered that a variable was related to a PC when $r \geq 0.6$. The PCA analyses and their visualization were performed with the packages FactoMineR and factoextra [64, 65]. All analyses were conducted in the program R version 3.4.2 [66].

## Results

### Diversity and composition of Cerambycidae beetles

A total of 1,323 individual saproxylic beetles were obtained from 57 species belonging to the family Cerambycidae (S1 Table). These emerged from 65 species of host trees from 28 families (S2 Table). Sample completeness was $\geq$ 97% ± 0.004% in all treatments (hereafter, we present the mean ± SD). Regarding $^0D$, no differences were found among treatments (Fig 1), while $^1D$ was greater in both dry season treatments (Dc, Dg), with no differences presented between these two treatments. However, there were differences between Rc and Rg, with the latter presenting greater $^1D$ diversity (Fig 1). The treatment that showed the greatest dominance ($^2D$) was Dg, with the least dominance found in Rc. No difference in species dominance was presented between Rg and Dc (Fig 1).

$\beta_{sor}$ showed a turnover of nearly half of the species between the four treatments, with values that ranged from 0.32 to 0.60 (0.48 ± 0.11). Beta diversity was mainly caused by the spatial arrangement of branches (ground vs. canopy) and the turnover of species ($\beta_{sim}$) took values

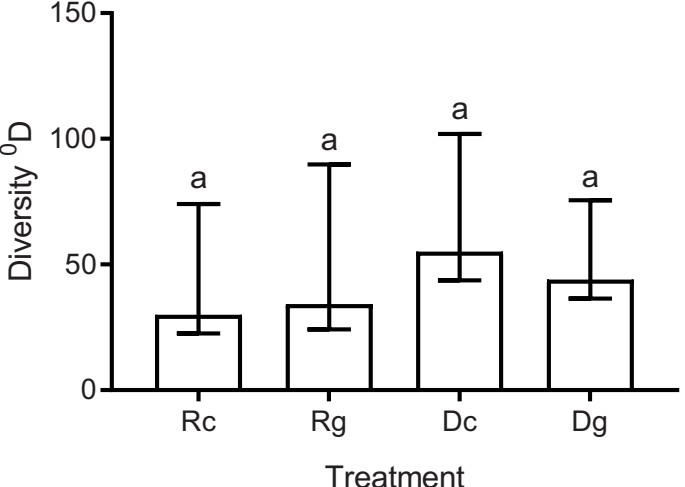

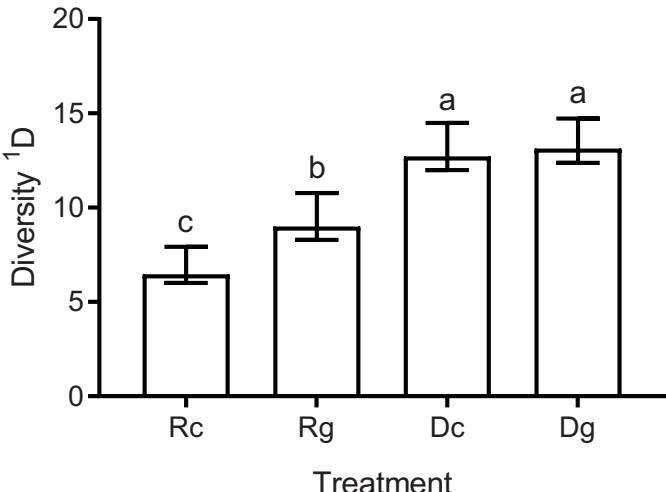

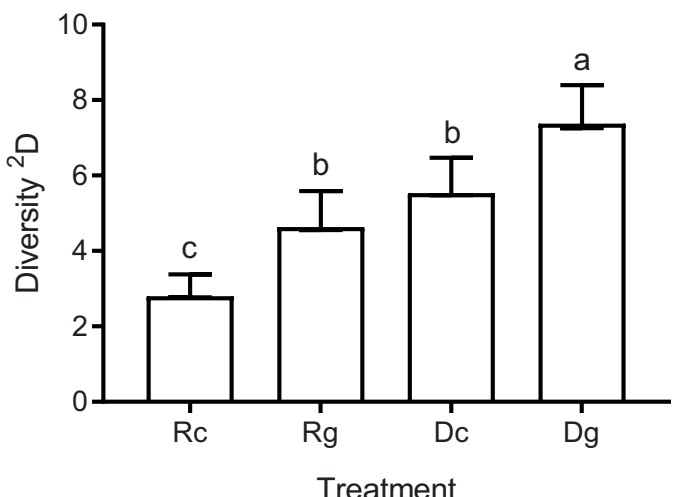

**Fig 1. Diversity of cerambycids ($^0$D, $^1$D, and $^2$D).** Diversity of cerambycids in the different treatments: Rain canopy (Rc), Rain ground (Rg), Dry canopy (Dc) and Dry ground (Dg) in a tropical dry forest. Mean values and 95% confidence intervals are presented. Different letters denote significant differences ($p < 0.05$) among treatments.

from 0.22 to 0.47 (0.36 ± 0.10; Fig 2A). Loss and/or gain of species ($\beta_{sne}$) contributed to a lesser extent to the turnover, since it presented values from 0.01 to 0.19 (0.12 ± 0.07; Fig 2A).

Analysis of interactions turnover ($\beta_{WN}$) showed high values within the four treatments, from 0.56 to 0.78 (0.71 ± 0.08; Fig 2B). The general pattern shows that the dissimilarity of interactions is due in equal magnitude to the turnover of species ($\beta_{ST}$), with values from 0.21 to 0.42 (0.35 ± 0.07), and the dissimilarity of interactions among shared species ($\beta_{OS}$; i.e. interaction rewiring), with values from 0.2 to 0.44 (0.35 ± 0.08). However, the turnover of interactions in the networks Rc→Rg was mainly due to the turnover of species ($\beta_{ST}$), while for the networks Dc→Dg it was interaction rewiring ($\beta_{OS}$) (Fig 2B).

## Structure and parameters of Cerambycidae-host tree interaction networks

The four networks comprised different numbers of host tree species and saproxylophagous beetles (Fig 3). The networks with the greatest number of interacting species were those of the dry season treatments (Dc, Dg; S1 Table). In general, we found interactions with softwood tree species and with species of cerambycids with lower body size (Fig 3).

The data showed that the Cerambycidae-host tree interaction networks are spatio-temporally stable and present a nested (9.08 ± 3.35) and modular (0.57 ± 0.07) structure (Table 1). Specialization ($H_2'$) was high in all of the networks, with values ranging from 0.64 to 0.72 (0.69 ± 0.03; Table 1). The niche overlap of the insects presented values from 0.04 to 0.06 (0.05 ± 0.01). For the host tree species, the values ranged from 0.16 to 0.25 (0.20 ± 0.04; Table 1).

On average, the interactions of the cerambycids are specialized, with values from 0.59 to 0.68 (0.64 ± 0.04). The effective partners for the insects presented values from 1.57 to 1.96 (1.77 ± 0.19). Species strength for the host species presented values from 0.78 to 0.95 (0.84 ± 0.07). The network parameters at species level among the treatments showed no significant differences [Beetles: specialization (d') H = 2.61; $p$ = 0.45; effective partners H = 1.60; $p$ = 0.65; Host tree: species strength H = 2.41; $p$ = 0.49].

## Trait-based predictors of network parameters

The PCA of the cerambycids of each treatment showed that > 78% of the variance was contained in two PC (S3 Table). In all of the treatments, PC 1 explained more than 50% of the variance and in the four networks the number of interactions correlated positively with the PC 1 (all $r$ > 0.97) and the effective partners (all $r$ > 0.97; S3 Table). PC 2 explained > 26% of the variance, but not all of the treatments had the same relationships. The PC 2 of the rainy season treatments (Rc, Rg) were related positively with d' (both $r$ > 0.72), while in Dc this correlation was of the same magnitude, but negative ($r$ = -0.78). The PC 2 of all of the treatments was related to body size, but with a different sign (all $r$ = 0.71 in Rg, Dc, Dg and -0.78 in Rc; S3 Table).

The general pattern in all of the treatments showed that, for the cerambycids, the PC 1 separated the saproxylophagous species *Eutrichillus comus* to the right side of the ordering. This is a generalist species that is very abundant in all of the treatments, for which it could be considered the most important species for the Cerambycidae-host tree interactions (S1 Fig). The PC 2 for the canopy treatments (Rc, Dc) showed a negative relationship between the specialization and the BSI. The upper part grouped the species with the greatest values of specialization (e.g.,

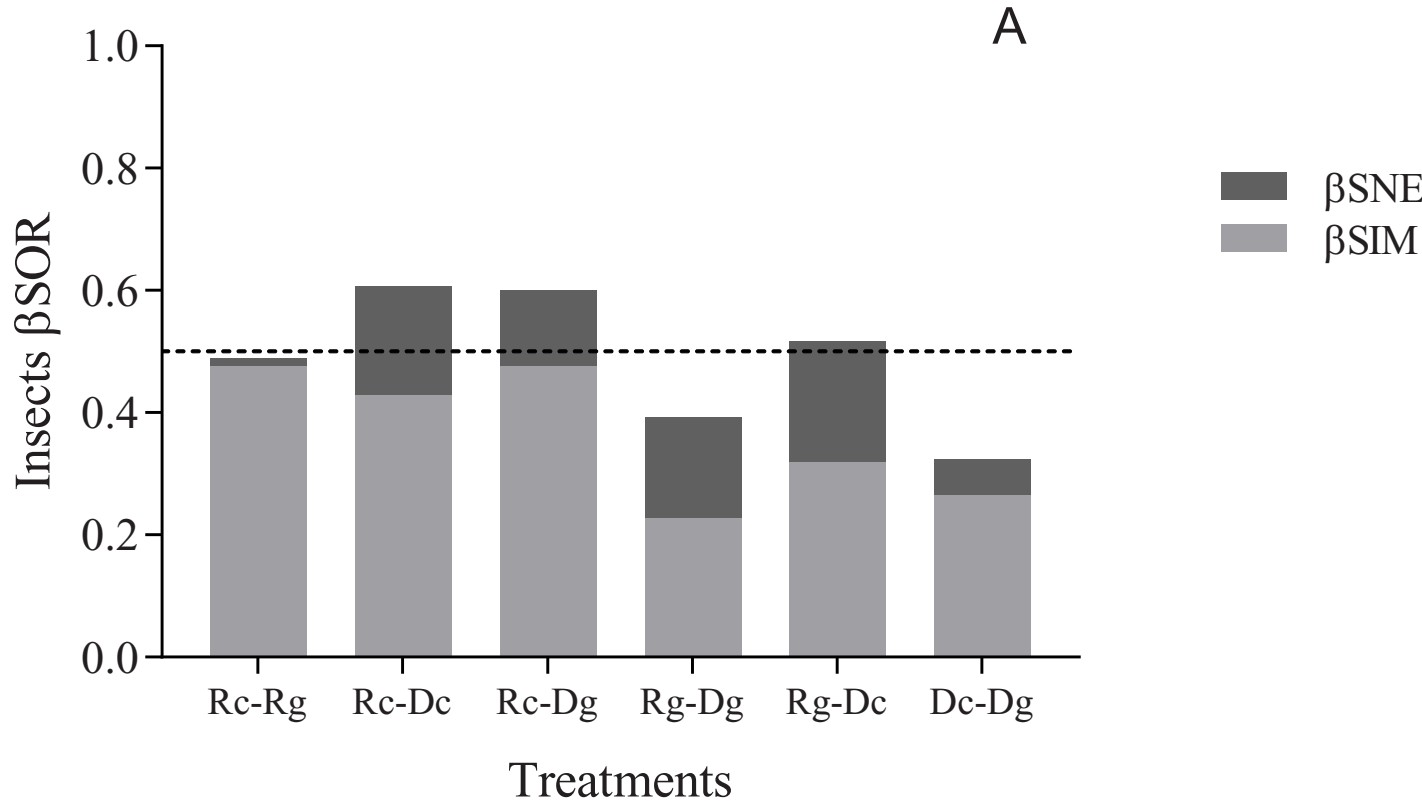

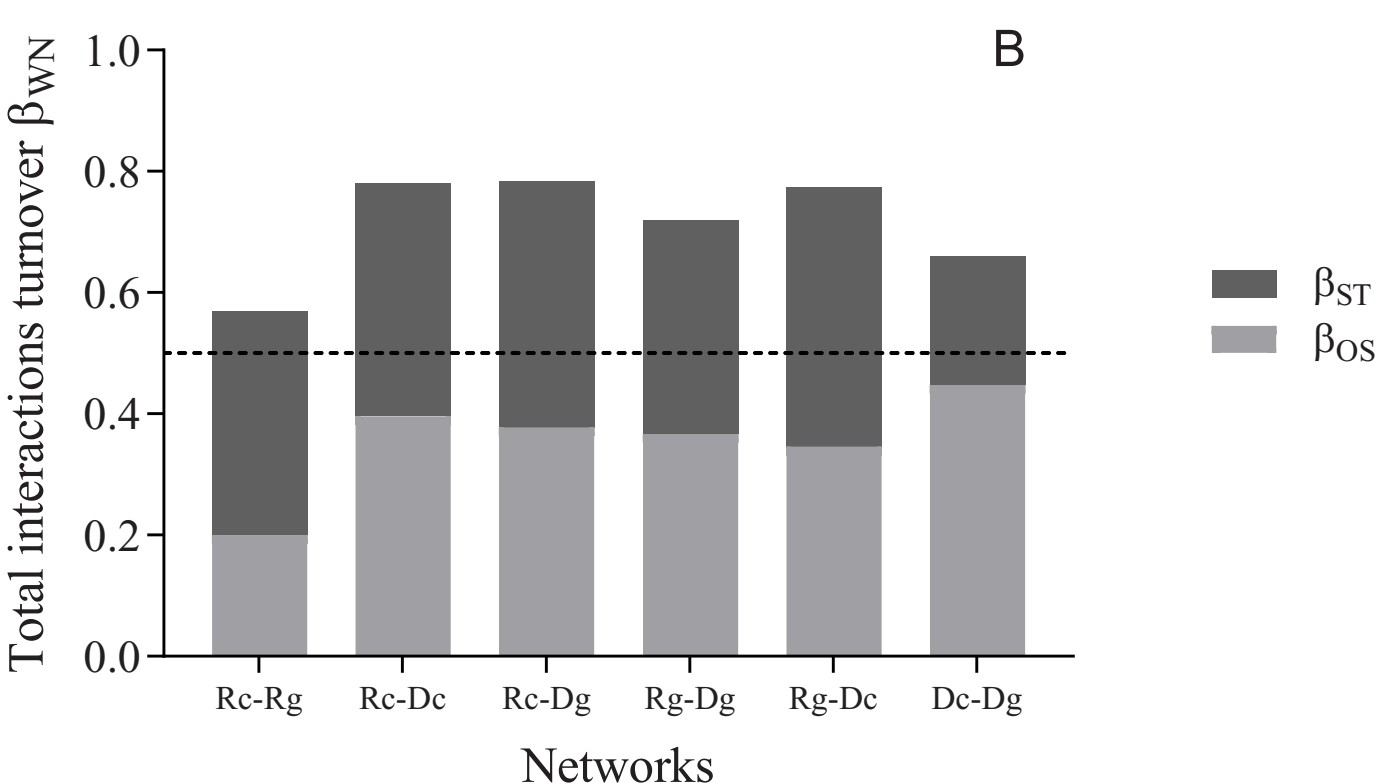

**Fig 2. Beta diversity of cerambycids.** Turnover of species and interactions (β-diversity) among treatments: Rain canopy (Rc), Rain ground (Rg), Dry canopy (Dc) and Dry ground (Dg) in a tropical dry forest. A) Sorensen dissimilarity of species ($\beta_{sor}$) and its components of turnover ($\beta_{sim}$) and nestedness ($\beta_{sne}$). B) Dissimilarity of interactions and its components ($\beta_{ST}$, dissimilarity in the interactions due to the similarity of the composition of species), and ($\beta_{OS}$, similarity of interactions in species that co-occur; i.e., species rewiring).

*Odontocera* sp., *Neocompsa puncticollis asperula*, *Euderces basimaculatus*) and with the highest BSI (e.g., *Neoptychodes trilineatus*, *Chyptodes dejeani*) (S1 Fig). For the treatment Rg, in the upper part are grouped the species with the highest body (e.g., *N. trilineatus*, *C. dejeani*). For the treatment Dc, the upper part grouped species with the highest BSI (e.g., *C. dejeani*, *Lagocheirus araneiformis ypsilon*) (S1 Fig).

The PCA of the host tree species showed that > 87% of the variance was contained in two PC (S4 Table). In all of the treatments, PC 1 explained > 56% of the variance and there was a positive correlation between the number of interactions (all $r > 0.81$) and species strength (all $r > 0.75$), but a negative correlation with wood hardness (all $r > -0.67$) and wood degradation (all $r > -0.65$; S4 Table). The PC 2 explained > 30% of the variance, but not all of the treatments presented the same relationships. The canopy treatments (Rc and Dc) were related positively with wood degradation ($r = 0.69$ and $0.66$, respectively) and only in Rg with wood hardness ($r = 0.62$); however, Dg in PC 2 was not related to any of the wood variables (S4 Table).

Regarding the general pattern of the PCA of the host tree species, PC 1 separated the plant species with a greater number of interactions and importance in the network (e.g., *B. copallifera*, *B. grandifolia*, *Conzattia multiflora*) to the right side of the ordering (S2 Fig). The PC 2 in the upper part of the canopy treatments (Rc and Dc) separated the host trees with the hardest wood (e.g., *Lysiloma divaricatum*, *Haematoxylum brasiletto*, *Diphysa robinioides*) and the treatment Rg with the greatest decomposition degree (e.g., *D. robinioides*, *Eysenhardtia polystachya*, *Wimmeria confusa*) (S2 Fig).

## Discussion

This is the first experimental study to analyze the spatio-temporal changes in diversity, composition and interaction patterns of the Cerambycid community with their host trees in a tropical dry forest. Our results confirm our hypotheses, since the network was specialized, regardless of space and time, but with a high turnover of species and interactions. Likewise, the volume and decomposition rate of the woods was related to their species strength and the BSI of the cerambycids to specialization at species level.

### Structure of Cerambycidae-host tree interaction networks

Our results show that the nested and modular structure of the Cerambycidae-host tree network remained spatio-temporally constant despite the high turnover of species and interactions among the treatments (see below). The nested pattern (WNODF = 9.08 ± 3.35) suggests that the interactions between cerambycids and their hosts can be generalists and specialist. Other studies have reported similar nestedness values in different habitat types (e.g., WNODF = 15, 12.12 and 8.36, [39]). Unlike other studies ([40] but see [38]), our results show a significantly modular structure; the modular network pattern is positively related with the specialization degree and suggests that groups of species co-evolved among themselves [57]. It is possible that the differences in the detection of the modular pattern among this and other studies (e.g., [40]), are due to the fact that previous studies use presence/absence metrics and in this study, we use quantitative analysis, which consider the intensity of the interaction (e.g., species abundance). However, the absence of modularity is to be expected in commensalistic

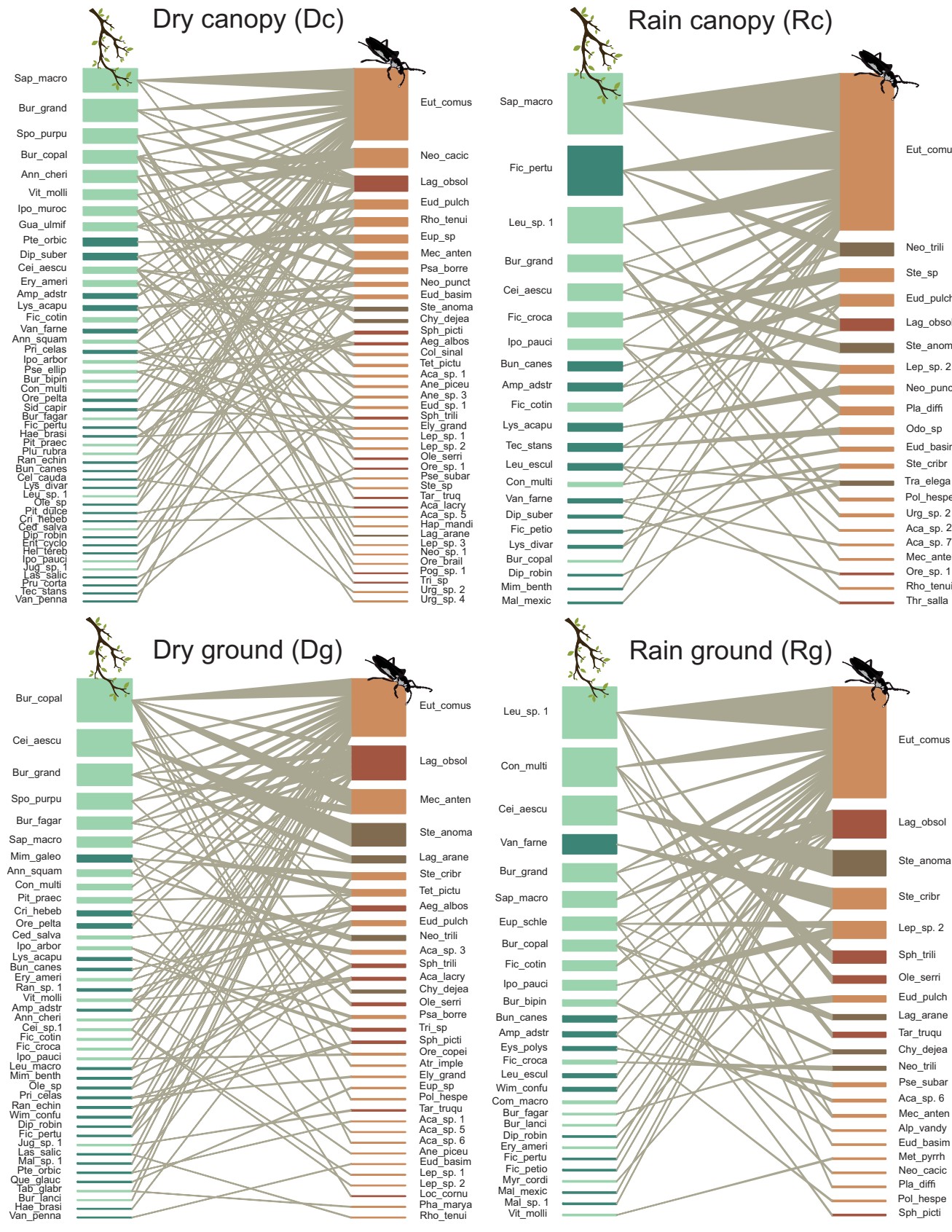

**Fig 3. Cerambycidae-host tree interaction networks of four treatments: Rain canopy (Rc), Rain ground (Rg), Dry canopy (Dc) and Dry ground (Dg) in a tropical dry forest.** The nodes of the Cerambycidae are represented by different brown tones: darker brown indicates a species with higher body size index and vice versa. Host tree nodes are represented by different green tones, darker green indicates tree species with harder wood and vice versa. The thickness of the links corresponds to the intensity of the interaction.

interactions, where interactions are asymmetrical and coevolution is lax, because at least for a group (e.g., trees) there are no direct benefits/costs associated with the use of its dead wood by cerambycids; while for the other group (i.e., cerambycids), a part of their life cycle depends on the availability of dead wood.

## Specialization of Cerambycidae-host tree interaction

Studies of host-specificity of Neotropical herbivores insects are based on plant phylogeny, suggest that there are a few species that feeding on a single plant species (i.e., specialist, [67]). However, in this experimental study performed with network analysis, we conclude in agreement with our hypothesis, that the network of commensalistic interactions between the cerambycids and their host trees was specialized ($H_2' = 0.69 \pm 0.03$), the difference may be due the difference in methods used. In agree with our results Wende et al. [25], report a specialized network ($H_2' = 0.76$) in a temperate forest in Germany, suggesting that, regardless of forest type, the cerambycids exhibit high selectivity for their hosts. It has been suggested that most of the insects tend to be specialists in host selection for oviposition since this selection has to ensure the development of the larvae in order to guarantee their adaptation (preference-performance hypothesis; [68]). Likewise, the saproxylophagous insects are adapted to feed on resources that change constantly in quality and contain defensive compounds [25], which could also influence the specialization of the network. In this sense, specialization as a result of diversification in resource use facilitates the coexistence of the species [69]. This agrees with the low values of niche overlap for the cerambycids ($0.05 \pm 0.01$), suggesting that, due to the specific requirements of each species, the cerambycids hardly share the niche in which they establish. It is important to highlight that the specialization is due to the selectivity of the cerambycids for tree species and not to a coevolutionary war, as in other systems (e.g., parasite-host). This selectivity of the cerambycids may be the result of competition between them or

**Table 1. Networks parameters.** Cerambycidae-host tree networks values in four treatments: Rain-canopy (Rc), Rain ground (Rg), Dry canopy (Dc) and Dry ground (Dg) in a tropical dry forest.

| | Treatments | | | |
|---|---|---|---|---|
| | **Rc** | **Rg** | **Dc** | **Dg** |
| Network size | 43 | 50 | 87 | 76 |
| Host tree species | 22 | 28 | 47 | 42 |
| Beetle species | 21 | 22 | 40 | 34 |
| **Structure** | | | | |
| Nestedness | 8.14* | 14.03* | 7.42* | 6.72* |
| Modularity | 0.47* | 0.62* | 0.61* | 0.56* |
| **Parameters** | | | | |
| $H_2$' | 0.71* | 0.72* | 0.70* | 0.64* |
| Niche overlap (T) | 0.21* | 0.25* | 0.16* | 0.16* |
| Niche overlap (B) | 0.06* | 0.06* | 0.05* | 0.04* |

T: Tree, B: Beetle.

* Denotes significant differences at $p < 0.01$

their inability to use all available woods (host limitation). In agreement with one of the commensalism assumptions (widespread use of interacting that does not face costs), the niche overlap of the hosts (0.20 ± 0.04) suggests that there are various tree species that can provide similar conditions for larvae of the same cerambycid species.

### Diversity and composition of species, and interaction turnover

Our hypothesis with respect to the changes in the diversity, composition, and interactions of the cerambycids was partially confirmed since we did not find differences in the species richness in any treatment. However, spatio-temporal changes occur with respect to the diversities $^1$D and $^2$D. This may be due to the fact that, in this study, there was a homogeneous quality and quantity of branches (resources) available (spatially and temporally) for the saproxylophagous beetles, causing the species richness to be constant, although this was not necessarily the case for the abundance of these species.

Diversity $^1$D (equally abundant species) was greater in the dry season treatments and lower in the treatment Rg. This result does not mean that there are more individuals in the dry season, it means that the proportion of the abundance among the species is more homogeneous (evenness) in the drought, increasing $^1$D, while the differences in abundance between the species are higher in rainy season. Reflecting the phenological behavior of the cerambycids, in the dry season, species with few individuals emerge that increase the evenness and make use of the few available resources. Further, the abundance of saproxylophagous beetles has been related to greater canopy openness, with greater availability of light and higher temperatures [20, 70], which could affect their metabolism and influence their diversity [71], particularly in those treatments conducted in the dry season. These conditions of higher temperatures modify the activity of the saproxylophagous insects [72], as well as their physiological processes (e.g., reducing the duration of the phase of development; [73]), for which reason it is possible that they act to promote their abundance. We found that the diversity $^2$D (equally dominant species) was greater in Dg and lower in Rc. One possible explanation for this pattern is that the treatment Dg was more exposed to conditions of higher temperatures, which could promote the development of the dominant species. Moreover, there is greater availability of dead wood in the TDF in the dry season [74], which would also contribute to the pattern of dominance observed.

In general, we found a spatio-temporal turnover of half of the species among treatments (except Rg-Dg 40% and Dc-Dg 35%), which is to be expected for insects of highly seasonal environments such as the TDF [74]. According to our findings, recently Martínez-Hernández et al., [75] found a complex relationship between the seasonal patterns in a TDF and the diversity and composition of the community of Cerambycidae. Also, Lee et al., [76] found a vertical stratification between seasons and forest stratum, suggesting that this pattern was occasioned for species of season and stratum generalists.

On analysis of the beta diversity, we found that the component that contributed most was the spatial turnover (36%), followed by the loss or gain of species (12%). This pattern could be explained by the prevailing microclimatic conditions in each of the treatments [77]. In this sense, the canopy species would be more exposed to greater variation in terms of solar radiation and humidity, while the conditions in the soil would be more stable.

In this study, we found a high turnover of interactions among the treatments (βWN; 71%), which was mainly due to the dissimilarity of interactions caused by the turnover of species (βST; 42%) and of shared species (βOS; 35%). This pattern shows that the interactions are not static and change through space and over time [78]. The spatio-temporal turnover between the cerambycids and their hosts could suggest a strategy to avoid competition over resources and to maintain the ensemble of these species in equilibrium [79].

## Trait-based predictors of network parameters

The indices related to the organization of the interactions (number of interactions and effective partners) explained the greater variation in the ordination (PC 1). It is possible that this pattern is due to generalist and abundant species such as *E. comus* and *Lagocheirus obsoletus obsoletus* [47, 80], which were the most important in terms of a number of interactions. Our hypothesis was partially confirmed, since the rest of the variation in the data (PC 2) was explained by the index of species-level specialization and the body size of the beetles in the canopy treatments (Rc, Dc). There is a non-linear relationship between host specificity, body size and, abundance in which the general tendency is that the largest insects are more specialized [25]. The exception in treatment Rg, where the relationship was positive, could be explained by the fact that we found species with a greater BSI (e.g., *C. dejeani*, *N. trilineatus*). This could be due to the fact that the soil or zones close to it are preferred by the saproxylophagous beetles because these offer more suitable conditions for their development [81], which concurs with the higher presence of equally abundant species and dominance of species in these treatments (see above).

In the host trees, the indices related to the organization and the importance of the species (i.e., number of interactions, species strength) explained the higher variation of the data (PC 1), which could be due to neutral processes, since the most important species are those of greatest abundance in the study site (e.g., *B. copallifera*, *B. grandifolia*, *C. multiflora*, *I. pauciflora*; [45].

On the other hand, our hypothesis related to the characteristics of the wood was confirmed, with the host species of softwood and those with a lower rate of decomposition proving to be the most important in terms of the interaction for the cerambycids. It is possible that tree species of rapid decomposition (e.g., *Tecoma stans*, *Pithecellobium dulce*), limit the cerambycids by not allowing their development and are ecological traps if females oviposit in them. Then, only species of cerambycids that have rapid development strategies can use the wood of these tree species. The rest of the variation of the data (PC 2) was generated by the hardness of the wood or the decomposition rate, which could be due to the fact that the most specialist host tree species (e.g., *Vachellia farnesiana*, *Lysiloma acapulcense*, *Mimosa galeottii*) were those with hardwoods. Nevertheless, some beetle species have evolved morphologically and physiologically in order to be able to develop in hardwoods [82]. In addition, the woods on which the saproxylophagous beetles feed in the initial phases of degradation still have several defensive substances (e.g., secondary metabolites, resins), which would promote more specialized interactions [83].

This study used the rearing of saproxylophagous insects which is the most suitable method for conducting studies of interactions with this group, since it avoids factors of confusion associated with the activity of the insects, as can occur with other methods (e.g., flight traps). For this reason, this experimental study in a TDF allowed us to evaluate the spatio-temporal changes in diversity, composition, and interactions of the community of saproxylophagous beetles associated with their host trees.

One of the great challenges of biological conservation is to predict how environmental and structural changes of the habitat can modify the species assemblage, and thus their interactions [84]. It is vitally important to conduct studies of conservation that consider the group of saproxylophagous beetles that arrive at the early stages of succession of the wood in decomposition since these insects are facilitators for other species as they act to promote infestation by fungi that are involved in the wood degradation processes [85]. Moreover, it is necessary to conserve the host species that comprise this group, particularly in the TDF, since these forests are under constant perturbation and are a reservoir of a great diversity of saproxylophagous insects [74].

## Supporting information

**S1 Fig. Beetle principal component analysis.**
(PDF)

**S2 Fig. Host tree principal component analysis.**
(PDF)

**S1 Table. Beetles species of tree-beetle network parameters per treatment in a tropical dry forest.** Degree (number of interactions), d (species specialization), Species strength (SS), Effective partners (EP) and a measure of body size index (BSI) presented the average.
(PDF)

**S2 Table. Tree host values of tree-beetle network parameters per treatment in a tropical dry forest.** Degree (number of interactions), d (species specialization), Species strength (SS), Effective partners (EP) and wood traits: Wood hardness (WH) and wood degradation (WD).
(PDF)

**S3 Table. Correlation values of four variables with the two main principal components.** A principal component analysis was performed for the insect species of the four treatments: Rain canopy (Rc), Rain ground (Rg), Dry canopy (Dc) and Dry ground (Dg) of tropical dry forest. In bold correlation values $r > 0.60$.
(PDF)

**S4 Table. Correlation values of four variables with the two main principal components.** A principal component analysis was performed for the tree host species of the four treatments: Rain canopy (Rc), Rain ground (Rg), Dry canopy (Dc) and Dry ground (Dg) of tropical dry forest. In bold correlation values $r > 0.60$.
(PDF)

## Acknowledgments

We would like to thank Gabriel Flores Franco for the assistance with species identification and HUMO and CIUM of the Universidad Autónoma del Estado de Morelos for allowing the comparison of specimens, for the determination of plants and cerambycids.

## Author Contributions

**Conceptualization:** Michelle Ramos-Robles, Orthon Ricardo Vargas-Cardoso, Angélica María Corona-López, Alejandro Flores-Palacios, Víctor Hugo Toledo-Hernández.

**Data curation:** Orthon Ricardo Vargas-Cardoso.

**Formal analysis:** Michelle Ramos-Robles, Alejandro Flores-Palacios, Víctor Hugo Toledo-Hernández.

**Investigation:** Alejandro Flores-Palacios, Víctor Hugo Toledo-Hernández.

**Methodology:** Michelle Ramos-Robles, Orthon Ricardo Vargas-Cardoso, Angélica María Corona-López, Víctor Hugo Toledo-Hernández.

**Project administration:** Orthon Ricardo Vargas-Cardoso.

**Software:** Michelle Ramos-Robles.

**Supervision:** Angélica María Corona-López, Víctor Hugo Toledo-Hernández.

**Validation:** Orthon Ricardo Vargas-Cardoso, Angélica María Corona-López, Alejandro Flo-res-Palacios, Víctor Hugo Toledo-Hernández.

**Writing – original draft:** Michelle Ramos-Robles, Angélica María Corona-López, Alejandro Flores-Palacios, Víctor Hugo Toledo-Hernández.

**Writing – review & editing:** Michelle Ramos-Robles, Angélica María Corona-López, Alejandro Flores-Palacios, Víctor Hugo Toledo-Hernández.

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
