## [Decision Letter · Decision Letter 0]

25 Sep 2019

PONE-D-19-19487

Spatio-temporal variation of Cerambycidae-host tree interaction networks

PLOS ONE

Dear Toledo-Hernández,

Thank you for submitting your manuscript to PLOS ONE. After careful consideration, we feel that it has merit but does not fully meet PLOS ONE’s publication criteria as it currently stands. Therefore, we invite you to submit a revised version of the manuscript that addresses the points raised during the review process.

This manuscript has now been evaluated by two reviewers. Both reviewers find important merits in the study, however they also raised important concerns, based on which we cannot accept this manuscript for publication at this time. Reviewer I is concerned about the analyses and their interpretation and think that the discussion should be improved at some parts. Reviewer II thinks that some parts of the methods and results need further clarification. Based on these evaluations, with which I concur, I would be happy to consider for further evaluation a substantially revised version of the manuscript that considers the suggestions and comments raised by the reviewers. If you decide to do this revision, please include also a cover letter indicating your responses to the reviewer comments and the changes you have made in the manuscript.  If you disagree with a reviewer's point, explain why.

We would appreciate receiving your revised manuscript by Nov 09 2019 11:59PM. To enhance the reproducibility of your results, we recommend that if applicable you deposit your laboratory protocols in protocols.io, where a protocol can be assigned its own identifier (DOI) such that it can be cited independently in the future. For instructions see: http://journals.plos.org/plosone/s/submission-guidelines#loc-laboratory-protocols

We look forward to receiving your revised manuscript.

Kind regards,

Amparo Lázaro, PhD

Academic Editor

PLOS ONE

Journal Requirements:

Reviewers' comments:

Reviewer's Responses to Questions

**Comments to the Author**

1. Is the manuscript technically sound, and do the data support the conclusions?

Reviewer #1: No

Reviewer #2: Yes

2. Has the statistical analysis been performed appropriately and rigorously? 

Reviewer #1: Yes

Reviewer #2: No

3. Have the authors made all data underlying the findings in their manuscript fully available?

Reviewer #1: Yes

Reviewer #2: Yes

4. Is the manuscript presented in an intelligible fashion and written in standard English?

Reviewer #1: Yes

Reviewer #2: Yes

5. Review Comments to the Author

Reviewer #1: In general, I like this paper as it brings data on the part of tropical forest food webs that is not often studied. The authors are right noting that there are very few studies of this kind, documenting tropical forest networks between trees and cerambycids. The data set is satisfactory in size, the analysis takes advantages of a number of new approaches to food web analysis. The authors also use the sensible wood traits to interpret host selection patterns by cerambycids. I particularly like the manipulative approach to the study of cerambycid communities, exposing standardized section of timber in different environments.

My only, but serious, concern here is that the authors, perhaps blinded by numerical methods of analysis, misinterpret their data by concluding that their networks are “with a high specialization degree” as stated in Abstract and repeated in the text. And yet, when you look at the actual food webs (Fig. 3) you can see that in all four food webs the most common cerambycid is always the same species, and that in all food webs it was reared from a large number of hosts, including always all three most used tree species. This pattern strongly suggests that the networks have actually very low degree of specialization, and that the high specialization values are caused by the predominance of rare species and interactions in the data, i.e. under-sampling. Every time a species is common, it feeds on numerous hosts. The authors also found “a high turnover of cerambycid species and their interactions” among the four food webs. However, these are food webs referring to spatially nearby sites (ground and canopy), and temporally adjacent seasons (dry and wet). For relatively mobile and long-living beetles such as cerambycids, such turnover in species and interactions is again unlikely to reflect biological reality, but more likely under-sampling.

Finally, the authors should better discuss the existing literature on cerambycid host specialization from the tropics. For instance, Tavakilian has provided some extensive data from (wet) Neotropics, and Novotny & basset (2005 Proc R Soc) review some more cerambycid data sets.

Reviewer #2: I have no doubt about the potential of the subject investigated by the authors. Congratulations on the excellent experimentally studied. I found the search extremely elegant and easy to read.

I hope my considerations will be helpful in improving the manuscript, and I appreciate the opportunity to review this manuscript.

Comments:

I really enjoyed reading the introduction and discussion of the manuscript. Both sections are well explained and concise, in my opinion. I congratulate the authors for the excellent text. My main questions are in the material and methods and the results. I made some considerations that deserve better care by the authors.

Lines 150-159: I may not have understood, but I would like to know how the host plants were collected. At each period (dry and rainy) were the trees sampled and made available for the beetles, or was only one collection performed for all experiments? Regardless of the way, it was not clear to me how the variation in host tree availability occurred (Table 1 in the results). Did all plant species chosen for the experiment have beetles emerged?

Lines 227-229: What does nestedness mean biologically for the system studied? The same goes for modularity analysis. It is not clear to me why there are two sections to talk about network level metrics (Structure of Cerambycidae-host tree interaction networks and Network level) whereas specialization, for example, is a network-level index such as nestedness and modularity.

Lines 231 and 234: Why did the authors use two different null models to test the significance of the analyzed metrics? How do these null models work? For example, Patefield algorithm is a restrictive null model. This is a good model to control for Type I error but leaves us exposed to Type II error. Please explain in the text why the authors are selecting this null model for modularity metric and null.t.test for the other metrics. In addition, the null.t.test is considered a very rough null-model test.

Lines 231-232: Regarding the modularity analysis you used the QuanBiMo algorithm. Since many methods for calculating modularity are now available, why did you choose this one? For example, Leger et al. (2015), Methods in Ecology and Evolution, 6:474-481 compared different methods and found that the Stochastic block models was the best method to retrieve modules in weighted networks. Authors may consider include a justification of the selected algorithm in Methods Section.

Even if the authors keep using an algorithm that maximizes modularity, because according to Leger et al. (2015) modularity maximization and one of the two variants of the Markov Chain Clustering algorithm (MCL1 / 10) were the models with the best performance after the stochastic block model, why didn't the authors calculate the recently implemented LPAwb + algorithm in the bipartite package? The LPAwb+algotithm robustly identify partitions with high modularity scores, showing to be efficient for the detection of subgroups in ecological networks (see Beckett 2016: https://doi.org/10.1098/rsos.140536).

Line 235: Which package was used to calculate network metrics?

Line 394: Note that nestedness means the existence of both generalist and specialist species in the networks.

Figure 3. The species names became too small, it might be interesting to change the abbreviated names to numbers for both beetles and plants.

6. PLOS authors have the option to publish the peer review history of their article (what does this mean?). If published, this will include your full peer review and any attached files.

Reviewer #1: No

Reviewer #2: No

---

## [Author Response · Author response to Decision Letter 0]

13 Dec 2019

The Response to Reviewers, Revised Manuscript with Track Changes and the unmarked version has been uploaded.

---

## [Decision Letter · Decision Letter 1]

27 Jan 2020

Spatio-temporal variation of Cerambycidae-host tree interaction networks

PONE-D-19-19487R1

Dear Dr. Toledo-Hernández,

We are pleased to inform you that your manuscript has been judged scientifically suitable for publication and will be formally accepted for publication once it complies with all outstanding technical requirements.

With kind regards,

Amparo Lázaro, PhD

Academic Editor

PLOS ONE

Additional Editor Comments (optional):

Reviewers' comments:

Reviewer's Responses to Questions

**Comments to the Author**

1. If the authors have adequately addressed your comments raised in a previous round of review and you feel that this manuscript is now acceptable for publication, you may indicate that here to bypass the “Comments to the Author” section, enter your conflict of interest statement in the “Confidential to Editor” section, and submit your "Accept" recommendation.

Reviewer #2: All comments have been addressed

2. Is the manuscript technically sound, and do the data support the conclusions?

Reviewer #2: Yes

3. Has the statistical analysis been performed appropriately and rigorously? 

Reviewer #2: Yes

4. Have the authors made all data underlying the findings in their manuscript fully available?

Reviewer #2: Yes

5. Is the manuscript presented in an intelligible fashion and written in standard English?

Reviewer #2: Yes

6. Review Comments to the Author

Reviewer #2: Dear Academic Editor,

Dr. Amparo Lázaro,

I have now reviewed this manuscript (Spatio-temporal variation of Cerambycidae-host tree interaction networks" - PONE-D-19-19487R1) by the second time. I have no doubt about the potential of the subject investigated by the authors. Congratulations on the excellent database collected during the field. Finally, I think the authors did a commendable job in improving it. All questions and suggestions were answered, and I found no further problems in the manuscript. They now provide more details regarding the procedures/new methods taken and the text has become clearer and easy to read. Congratulations on the work done, and I recommend publishing.

7. PLOS authors have the option to publish the peer review history of their article (what does this mean?). If published, this will include your full peer review and any attached files.

Reviewer #2: No

---

## [Editor Report · Acceptance letter]

3 Feb 2020

PONE-D-19-19487R1 

Spatio-temporal variation of Cerambycidae-host tree interaction networks 

Dear Dr. Toledo-Hernández:

I am pleased to inform you that your manuscript has been deemed suitable for publication in PLOS ONE. Congratulations! Your manuscript is now with our production department. 

With kind regards,

on behalf of

Dr. Amparo Lázaro 

Academic Editor

PLOS ONE